# Distribution-dependent and Time-uniform Bounds for Piecewise i.i.d Bandits

**Subhojyoti Mukherjee** [1]   **Odalric Maillard** [2]

## Abstract

We consider the setup of stochastic multi-armed bandits in the case when reward distributions are piecewise i.i.d. and bounded with unknown changepoints. We focus on the case when changes happen simultaneously on all arms, and in stark contrast with the existing literature, we target gap-dependent (as opposed to only gap-independent) regret bounds involving the magnitude of changes ($\Delta_{i,g}^{chg}$) and optimality-gaps ($\Delta_{i,g}^{opt}$). Under a slightly stronger set of assumptions, we show that as long as the compounded delayed detection for each changepoint is bounded there is no need for forced exploration to actively detect changepoints. We introduce two adaptations of UCB-strategies that employ scan-statistics in order to actively detect the changepoints, without knowing in advance the changepoints and also the mean before and after any change. Our first method UCBL-CPD does not know the number of changepoints $G$ or time horizon $T$ and achieves the first time-uniform concentration bound for this setting using the Laplace method of integration. The second strategy ImpCPD makes use of the knowledge of $T$ to achieve the order optimal regret bound

of $\qquad \min\big\{O(\sum_{i=1}^{K}\sum_{g=1}^{G}\frac{\log(T/H_{1,g})}{\Delta_{i,g}^{opt}}), O(\sqrt{GT})\big\}$,

(where $H_{1,g}$ is the problem complexity) thereby closing an important gap with respect to the lower bound in a specific setting. Our theoretical findings are supported by numerical experiments on synthetic and real-life datasets.

## 1. Introduction

In this paper, we consider the piecewise i.i.d multi-armed bandit problem, an interesting variation of the stochastic multi-armed bandit (SMAB) setting. The learning algorithm is provided with a set of decisions (or arms) which belong to the finite set $\mathcal{A}$ with individual arm indexed by $i$ such that $i = 1, \ldots, K$. The learning proceeds in an iterative fashion, where in each time step $t$, the algorithm chooses an arm $i \in \mathcal{A}$ and receives a stochastic reward that is drawn from a distribution specific to the arm selected. There exist a finite number of changepoints $G$ such that the reward distribution of all arms changes at those changepoints. $\mathcal{G}$ denotes the finite set of changepoints indexed by $g = 1, \ldots, G$, and $t_g$ denotes the time step when the changepoint $g$ occurs. At $\tau_g$ the learner detects $t_g$ with some delay. The reward distributions of all arms are unknown to the learner. The learner has the goal of maximizing the cumulative reward at the end of the horizon $T$.

The piecewise i.i.d setting is extremely relevant to a lot of practical areas such as recommender systems, industrial manufacturing, and medical applications. In the health-care domain, the non-stationary assumption is more realistic than i.i.d. assumption, and thus progress in this direction is important. An interesting use-case of this setting arises in drug-testing for a cure against a resistant bacteria, or virus such as AIDS. Here, the arms can be considered as various treatments while the feedback can be considered as to how the bacteria/virus reacts to the treatment administered. It is common in this setting that the behavior of the bacteria/virus changes after some time and thus its response to all the arms also change simultaneously. Moreover, in the health-care domain it is extremely risky to conduct additional uniform exploration of actions to detect simultaneous abrupt changes and relying only on the past history of interactions might lead to reliable detection of changepoints and safer policies.

A key aspect of this setting that the previous approaches have not fully utilized is that in the global changepoint setup when gaps are large enough on each arm so that they can be detected long before the next changepoint, *no additional forced exploration* is required to actively detect change-points (like CUSUM (Liu et al., 2017), M-UCB (Cao et al., 2018)). But, this has to be done carefully as detecting a changepoint at $t_g$ requires some delay ($\tau_g$) and this might

[*]Equal contribution [1]College of Information & Computer Science, University of Massachusetts Amherst, Amherst, USA [2]SequeL team, INRIA Lille - Nord Europe, France. Correspondence to: Subhojyoti Mukherjee <subho@cs.umass.edu>, Odalric Maillard <odalric.maillard@inria.fr>.

*Reinforcement Learning for Real Life (RL4RealLife) Workshop in the $36^{th}$ International Conference on Machine Learning*, Long Beach, California, USA, 2019. Copyright 2019 by the author(s).

endanger the detection of the next changepoint. This problem of *compounded delayed detection* becomes trickier if the successive changepoint gaps are not large enough and if they are not well separated. To avoid these complexities, past approaches have relied on just minimizing regret over a short window (passive algorithms) or conduct additional exploration regularly to detect changepoints and reset (active algorithms).These also inherently lead to less powerful and less informative worst case gap-independent bounds that depend on optimality gaps implicitly by assuming that gaps are evolving at a constant frequency, thus requiring the knowledge of $G$ or $T$ or both.

However, we can do better by utilizing three mild assumptions and reduce the regret incurred for forced explorations. These are assumption on global changepoint (1), a different separation of changepoint assumption (2) based on maximal delay, and a slightly stronger assumption (3) involving minimum detectable gaps ($\Delta(t_g, \delta)$). All the previous algorithms take some combination of these assumptions (see Table 1). But these only partially solve our problems, as specific care still needs to be taken by individual strategies as we have to show for each of our proposed detection strategies that their compounded delayed detection happens well before the next changepoint sets in. This approach provides the opportunity to study this interesting setting as an extension of SMAB. Hence, we can prove the gap-dependent regret upper bound that consist of both the changepoint gaps ($\Delta_{i,g}^{chg}$) and optimality gaps ($\Delta_{i,g}^{opt}$) for each changepoint $g \in G$. Furthermore, now we will be able to give minimax (see 2.4.3 in Bubeck & Cesa-Bianchi (2012)) regret bounds that incorporates the *Hardness factor* $H_{1,g}$ (introduced in Audibert et al. (2010), best-arm identification) which characterizes how difficult is the environment and depends on both $\Delta_{i,g}^{chg}$ and $\Delta_{i,g}^{opt}$. Also, for this setting we will be needing an additional hardness parameter $H_{2,g}$ which captures the trade-off between optimality and changepoint gaps. We conjecture that an order optimal regret upper bound in the piecewise i.i.d setting of the order

$$\min\left\{ O\left( \sum_{g=1}^{G} \sum_{i=1}^{K} \frac{H_{2,g} \log(T/(GH_{1,g}))}{\Delta_{i,g}^{opt}} \right), O\left(\sqrt{GT}\right) \right\}$$

is attainable. Obtaining such optimal minimax bound for SMAB was discussed in Audibert & Bubeck (2009), Auer & Ortner (2010), Bubeck & Cesa-Bianchi (2012) and solved in Lattimore (2016). We further extend the results to piecewise i.i.d algorithms which is non-trivial given the changepoint and optimality gaps have to be tackled independently. Our contributions are mainly threefold: *Algorithmic, Time-Uniform bound*, and *Order-optimal Regret bound*.

**1. Algorithmic:** We propose two actively adaptive upper confidence bound (UCB) algorithms, referred to as UCBLaplace-Changepoint Detector (UCBL-CPD ) and Im-

*Table 1.* Comparison of Algorithms

| Algorithm | Type | T | G | Assumptions |
|---|---|---|---|---|
| ImpCPD (ours) | Active | Y | N | 1, 2, 3 |
| UCBL-CPD (ours) | Active | N | N | 1, 2, 3 |
| CUSUM | Active | Y | Y | 1, 2[1], 3 |
| M-UCB | Active | Y | Y | 1, 2[1], 3 |
| EXP3.R | Active | Y | N | 3 |
| DUCB | Passive | Y | Y | 1, 3 |
| SWUCB | Passive | Y | Y | 1, 3 |
| Lower Bound | Oracle | Y | Y | 1, 3 |

proved Changepoint Detector (ImpCPD ). Unlike CD-UCB, M-UCB and CUSUM, UCBL-CPD and ImpCPD do not conduct forced exploration to detect changepoints. They divide the time into slices, and for each time slice, for every arm they check the UCB, LCB mismatch based on past observations only to detect changepoints. While UCBL-CPD checks for every such combination of time slices, ImpCPD only checks at certain estimated points in time horizon, and hence saves on computation time. Thus, controlling the compounded detection delay for ImpCPD is trickier than UCBL-CPD .

**2. Time-uniform bound:** The previous approaches CDUCB, CUSUM uses the Chernoff-Hoeffding inequality and M-UCB uses McDiarmids inequality with union bound to obtain the regret bound. DUCB and SWUCB uses the peeling argument which results in slightly tighter concentration bound but both these techniques results in less tight bounds than the Laplace method of integration used for UCBL-CPD . Moreover, UCBL-CPD has time-uniform bound as its confidence interval does not depend explicitly on $t$ as opposed to other methods. On the contrary, ImpCPD which is not anytime and has access to $T$ uses the usual union bound with geometrically increasing phase length (like peeling) is somewhat in between union and peeling argument. Both these proofs are of independent interest which can be used in other settings as well. A detailed comparison of union, peeling and Laplace bound can be found in Discussion 2.

**3. Regret bound:** We prove the gap-dependent regret upper bound that consist of both the changepoint gaps ($\Delta_{i,g}^{chg}$) and optimality gaps ($\Delta_{i,g}^{opt}$) for each changepoint $g \in \mathcal{G}$, $i \in \mathcal{A}$ in Theorem 1, and 2 (under assumptions 1, 2, and 3). For the gap-independent result we show in the special case when all the gaps are same such that for all $i \in \mathcal{A}, g \in \mathcal{G}$, $\Delta_{i,g}^{opt} = \Delta_{i,g}^{chg} = \Delta(t_g, \delta)$, $H_{1,g} = K(\Delta(t_g, \delta))^{-2}$ and $H_{2,g} = 1$, UCBL-CPD and ImpCPD achieves $\sqrt{GT} \log T$ and $\sqrt{GT}$ respectively (Table 2, Corollary 1, and 2). Such a setup when all the gaps scale atleast as $\Omega(\sqrt{1/T})$ is natural in view of existing works like Audibert & Bubeck (2009),

---

[1]Requires a different separation of changepoint assumption.

*Table 2.* Regret Bound of Algorithms

| Algorithm | Gap-Dependent | Gap-Independent |
|---|---|---|
| ImpCPD (ours) | Theorem 2 | $O\left(\sqrt{GT}\right)$ |
| UCBL-CPD (ours) | Theorem 1 | $O\left(\sqrt{GT}\log T\right)$ |
| CUSUM | N/A | $O\left(\sqrt{GT\log\frac{T}{G}}\right)$ |
| M-UCB | N/A | $O\left(\sqrt{GT\log T}\right)$ |
| EXP3.R | N/A | $O\left(G\sqrt{T\log T}\right)$ |
| DUCB | N/A | $O\left(\sqrt{GT}\log T\right)$ |
| SWUCB | N/A | $O\left(\sqrt{GT\log T}\right)$ |
| Lower Bound | Theorem 3 | $\Omega\left(\sqrt{GT}\right)$ |

Auer & Ortner (2010). We also prove a gap-dependent and independent lower bound for an optimal oracle policy $\pi^*$ as opposed to Garivier & Moulines (2011) which proves a gap-independent lower bound for a policy $\pi$ only when $G = 2$ (Theorem 3). Note, that an oracle policy $\pi^*$ has access to the exact changepoints, where it is restarted with its past history of interactions erased. We show that UCBL-CPD and ImpCPD perform very well across diverse piecewise i.i.d environments (Section 6, and Appendix N).

The rest of the paper is organized as follows. We first setup the problem in Section 2. Then we define our problem statement in Section 3, and in Section 4 we present the changepoint detection algorithms. Section 5 contains our main result, remarks and discussions. Section 6 contains numerical simulations, Section 7 contains related works, and we conclude in Section 8. The proofs are provided in Appendices in the supplementary material. The supplementary material is provided in this link. A full updated version of the paper is available in this link.

## 2. Preliminaries

We assume all rewards are bounded in $[0, 1]$. $N_{i,\tau_g:\tau_{g+1}-1}$ denotes the number of times arm $i$ has been pulled between $\tau_g$ to $\tau_{g+1} - 1$ timesteps for any sequence of increasing $(\tau_g)_g$ of integers. Also, we define $\hat{\mu}_{i,\tau_g:\tau_{g+1}-1}$ as the empirical mean of the arm $i$ between $\tau_g$ to $\tau_{g+1} - 1$ timesteps. We consider that on each arm i, the process generating the reward is piecewise mean constant according to the sequence $(t_g)_g$. That is, if $\mu_{i,t}$ denotes the mean reward of arm i at time t, then $\mu_{i,t}$ has same value for all $t \in \rho_g$.

**Definition 1.** *We define $t_g$ as,*

$$t_g = \min\{t > t_{g-1} : \exists i, \mu_{i,t-1} \neq \mu_{i,t}\}$$

**Assumption 1.** *(Global changepoint) We assume the global changepoint setting, that is $t_g = t$ implies $\mu_{i_{t-1}} \neq \mu_{i_t}$, for all $i \in \mathcal{A}$.*

**Definition 2.** *Let the changepoint gap at $t_g$ for an arm*

$i \in \mathcal{A}$ *between the segments $\rho_g$ and $\rho_{g+1}$ be denoted as,*

$$\Delta_{i,g}^{chg} = |\mu_{i,g} - \mu_{i,g+1}|.$$

Thus, at each changepoint we assume that the mean of all the arms change. So our assumption is stricter than Liu et al. (2017), Cao et al. (2018), Besson & Kaufmann (2019) where at $t_g$, $\mu_{i_t}$ of any arm may or may not change requiring the forced exploration of all arms to detect changepoint. We make a distinction between the finite set of all arms $\mathcal{A}$ and $\mathcal{A}_g^{chg}$, such that $\mathcal{A}_g^{chg}$ denotes only those arms $i \in \mathcal{A}$ whose $\Delta_{i,g}^{chg} > 0$ at the $t_g$-th changepoint.

Now, we carefully setup our problem. First, we define the minimum number of samples $n(t_g, \Delta, \delta)$ required for an arm $i \in \mathcal{A}$ so that a deviation of $\Delta > 0$ of $\hat{\mu}_{i,g-1}$ (or $\hat{\mu}_{i,g}$) from $\mu_{i,g-1}$ (or $\mu_{i,g}$) before and after the $t_g$-th changepoint can be controlled with $(1 - \delta)$ probability. This is shown in Lemma 1. We then define the notion of maximal delay of a best policy $\pi^*$ in detecting a changepoint at $t_g$ starting exactly from $t_{g-1}$ (Lemma 2). Finally, we define the detectable changepoint gap $\Delta(t_g, \delta)$ based on the maximal delay of $\pi^*$ and $n(t_g, \Delta, \delta)$.

**Lemma 1.** *(Control of large deviations) For our detection strategy using estimated means, it is sufficient to collect a minimum number of samples $n(t_g, \Delta, \delta) = \lceil \frac{1}{2}\log(\frac{2(t-t_{g-1})^2}{\delta})/\Delta^2 \rceil$ for an arm $i \in \mathcal{A}_g^{chg}$ before or after $t_g$ so that $|\hat{\mu}_{i,g-1} - \mu_{i,g-1}| \leq \Delta$ or $|\hat{\mu}_{i,g} - \mu_{i,g}| \leq \Delta$ with $(1 - \delta)$ probability and $t_{g-1} < t_g < t$.*

**Proof 1.** The proof of Lemma 1 is given in Appendix B.

**Assumption 2.** *(Separation of Changepoints) We assume that for every two consecutive segments $\rho_{g-1}$ and $\rho_g$, $\forall g = 1, 2, \ldots, G - 1$ all the three changepoints $t_{g-1}, t_g$ and $t_{g+1}$ satisfy the following condition,*

$$t_g + d_{\pi^*}(t_g - t_{g-1}) \leq t_g + \eta(t_{g+1} - t_g) = \eta t_{g+1} + (1 - \eta)t_g$$

*where $\eta \in (0, 1)$, and $d_{\pi^*}(t_g - t_{g-1})$ is the maximal delay of a best detection strategy starting at $t_{g-1}$.*

Using Lemma 1 and Assumption 2 now we properly define detectable changepoint gap $\Delta(t_g, \delta)$ to be such $\Delta(t_g, \delta) \geq \sqrt{\log(2(x^2/\delta))/2x}$ where $x = t_g + d_{\pi^*}(t_g - t_{g-1}) - t_{g-1}$.

**Lemma 2.** *(Detection Delay) With the standard assumption that at $t_g$, $\Delta(t_g, \delta)$ scales atleast as $\Omega(\sqrt{\frac{\log t}{t}})$ for K arms, then to detect a deviation of $\Delta \geq \Delta(t_g, \delta)$ with probability $(1 - \delta)$, there exists a best detection strategy starting at $t_{g-1}$ that suffers a worst case maximum delay of,*

$$d_{\pi^*}(t_g - t_{g-1}) \leq \left(\frac{C(t,\delta,\eta)K\log(\frac{t^2}{\delta})}{2(\Delta(t_g,\delta))^2}\right) + K\delta$$

*where, $C(t, \delta, \eta) \leq \eta\log(t/\delta)$, and $\eta \in (0, 1)$.*

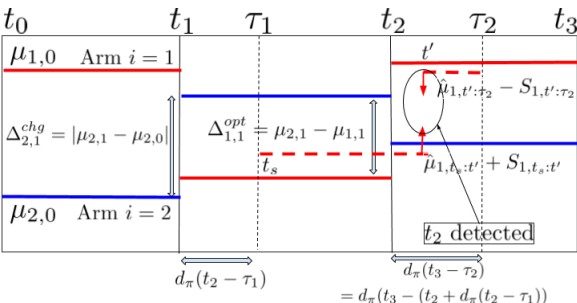

*Figure 1.* A 2-arm and 2 changepoint scenario

**Definition 3.** *The changepoint gap $\Delta_{i,g}^{chg}$ for an arm $i \in \mathcal{A}_g^{chg}, g \in \mathcal{G}$ is a $\delta$-optimal changepoint gap if, $\Delta_{i,g}^{chg} \geq \Delta(t_g, \delta)$.*

**Assumption 3.** *(Minimum gap)* We assume that $\forall g \in \mathcal{G}$ the changepoint gaps $\Delta_{i,g}^{chg}, \forall i \in \mathcal{A}$ are $\delta$-optimal gaps.

**Discussion 1.** Thus, Assumption 2 makes sure that $t_g + d_{\pi^*}(t_g - t_{g-1})$ stays away from $t_{g+1}$. This ensures that when restarting the detection strategy from $t_g + d_{\pi^*}(t_g - t_{g-1})$, the detection of $t_{g+1}$ will not be too much endangered. Moreover, all the gaps $\Delta_{i,g}^{chg}, i \in \mathcal{A}_g^{chg}, \forall g \in \mathcal{G}$ are $\delta$-optimal changepoint gaps following Assumption 3. A $\delta$-optimal changepoint gap requires a minimum sample of $n(t_g, \Delta, \delta)$ to ensure that $\Delta \geq \Delta(t_g, \delta)$ deviation from the mean occur with $(1 - \delta)$ probability (Lemma 1). A reasonable detection strategy $\pi$ has its maximal delay $d_\pi(t_g - \tau_{g-1}) \geq d_{\pi^*}(t_g - t_{g-1})$ as it only has observation to detect $t_g$ from $\tau_{g-1}$ and not from $t_{g-1}$ (unlike $\pi^*$). It tries to minimize $d_\pi(t_g - \tau_{g-1})$ in detecting $t_g$ such that $d_\pi(t_g - \tau_{g-1}) \leq (1 + \beta)d_{\pi^*}(t_g - t_{g-1})$ holds with high probability for some $\beta \in (0,1)$ and $\eta \in (0,1)$. This approach is different than Cao et al. (2018), Liu et al. (2017), Besson & Kaufmann (2019) as they assume that all the previous changepoints before $t_g$ has been detected which is unrealistic. In contrast, we assume that for a changepoint $t_g$ if the number of available observations are large enough then the $t_g$-th changepoint is detected irrespective of the detection of other changepoints (given assumption 2 and 3 holds). An illustrative explanation is shown in Figure 1.

**Definition 4.** *Let the optimality gap $\Delta_{i,g}^{opt}$ for an arm $i_{t'} \neq i_{t'}^*, \forall t' \in [t_{g-1}, t_g - 1]$ be defined as, $\Delta_{i,g}^{opt} = \mu_{i^*,g} - \mu_{i,g}$.*

## 3. Problem Formulation

**Regret Definition:** The objective of the learner is to minimize the cumulative regret till $T$, which is defined as:

$$R_T = \sum_{t'=1}^{T} \mu_{i_{t'}^*} - \sum_{t'=1}^{T} \mu_{i_{t'}} \mathbb{I}\{i_{t'} \neq i_{t'}^*\}$$

where $T$ is the horizon, $\mu_{i_{t'}^*}$ is the expected mean of the optimal arm at the $t'$ timestep and $\mu_{i_{t'}} \mathbb{I}\{i_{t'} \neq i_{t'}^*\}$ is the

expected mean of the arm chosen by the learner at the $t'$ timestep when it was not the optimal arm $i_{t'}^*$. The expected regret of an algorithm after $T$ rounds can be written as,

$$\mathbb{E}[R_T] = \mathbb{E}\left[\sum_{t'=1}^{T} \mu_{i_{t'}^*} - \sum_{t'=1}^{T} \mu_{i_{t'}} \mathbb{I}\{i_{t'} \neq i_{t'}^*\}\right]$$

$$\stackrel{(a)}{=} \mathbb{E}\left[\sum_{g=1}^{G} \sum_{t'=t_{g-1}}^{t_g} \mu_{i_{t'}^*} - \sum_{g=1}^{G} \sum_{t'=t_{g-1}}^{t_g} \mu_{i_{t'}} \mathbb{I}\{i_{t'} \neq i_{t'}^*\}\right]$$

$$\stackrel{(b)}{=} \sum_{g=1}^{G} \sum_{i=1}^{K} \Delta_{i,g}^{opt} \mathbb{E}[N_{i,t_{g-1}:t_g}]$$

where $(a)$ is from Assumption 1, and $(b)$ from Definition 4.

**Problem Complexity:** We define the hardness of a changepoint $g \in \mathcal{G}$ using optimality and changepoint gaps by modifying the definitions of *problem complexity* as introduced in Audibert et al. (2010) for stochastic bandits:

$$H_{1,g} = \max\left\{\sum_{i=1}^{K} \frac{1}{(\Delta_{i,g}^{opt})^2}, \sum_{i \in \mathcal{A}_g^{chg}} \frac{1}{(\Delta_{i,g}^{chg})^2}\right\}, H_{2,g} = \frac{\Delta_{\max,g+1}^{opt}}{\Delta(t_g, \delta)}$$

where, $\Delta_{\max,g+1}^{opt} = \max_{i \in \mathcal{A}} \Delta_{i,g+1}^{opt}$. The hardness parameter $H_{2,g}$ captures the tradeoff between the minimum detectable gap $\Delta(t_g, \delta)$ and maximum optimality gap of the next changepoint $\Delta_{\max,g+1}^{opt}$ which serves as an upper bound to all such possible trade-offs at changepoint $g$. The relation between the above complexity terms can be derived as:

$$H_{2,g} \leq H_{1,g} \leq \frac{K}{(\Delta(t_g, \delta))^2}(H_{2,g}).$$

Note that, $\frac{\Delta_{i,g}^{opt}}{\Delta_{i,g}^{chg}} \leq H_{2,g}, \forall i \in \mathcal{A}_g^{chg}, g \in \mathcal{G}$. In the special case when $\Delta_{i,g}^{opt} = \Delta_{i,g}^{chg} = \Delta(t_g, \delta), \forall i \in \mathcal{A}_g^{chg}, g \in \mathcal{G}$, then,

$$H_{1,g} = K(\Delta(t_g, \delta))^{-2}, \text{ and } H_{2,g} = 1$$

## 4. Algorithms

We first introduce the policy UCBL-CPD in Algorithm 1 which is an adaptive algorithm based on the standard UCB1 (Auer et al., 2002a) approach. UCBL-CPD pulls an arm at every timestep as like UCB1 but has the time-uniform concentration bound that holds simultaneously for all timestep $t$. It calls upon the Changepoint Detection (CPD) subroutine in Algorithm 2 for detecting a changepoint. Note, that unlike CD-UCB, CUSUM, and M-UCB the UCBL-CPD does not conduct forced exploration to detect changepoints. UCBL-CPD is an anytime algorithm which does not require the horizon as an input parameter or to tune its parameter $\delta$. This is in stark contrast with CD-UCB, CUSUM, M-UCB, DUCB or SWUCB, that require the knowledge of $G$ or $T$

for optimal performance. we define the confidence interval of UCB-Laplace as follows:

$$S_{i,t_s:t_p} := \sqrt{\left(1 + \frac{1}{N_{i,t_s:t_p}}\right) \frac{\log(\sqrt{N_{i,t_s:t_p} + 1/\delta})}{2N_{i,t_s:t_p}}} \quad (1)$$

---

**Algorithm 1** UCB Laplace CPD (UCBL-CPD )

---

1: **Input:** $\delta > 0$;
2: **Definition:** $S_{i,t_s:tp}$ from (1)
3: **Initialization:** $t_s := 1$, $t_p := 1$.
4: Pull each arm once
5: **for** $t = K + 1, .., T$ **do**
6:      Pull arm $j \in \arg\max_{i\in\mathcal{A}} \left\{ \hat{\mu}_{i,t_s:t_p} + S_{i,t_s:t_p} \right\}$, observe reward $X_{j,t}$.
7:      Update $\hat{\mu}_{j,t_s:t_p}$, $N_{j,t_s:t_p} := N_{j,t_s:t_p} + 1$.
8:      $t_p := t_p + 1$.
9:      **if** (CPD($t_s, t_p, \delta$)) **then**
10:         **Restart:** Set $\hat{\mu}_{i,t_s:t_p} := 0$, $N_{i,t_s:t_p} := 0, \forall i \in \mathcal{A}$, $t_s := t$, $t_p := t_s$.
11:         Pull each arm once.
12:      **end if**
13: **end for**

---

**Algorithm 2** Changepoint Detection($t_s, t_p, \delta$) (CPD)

---

1: **for** $i = 1, .., K$ **do**
2:      **for** $t' = t_s, .., t_p$ **do**
3:         **if** $\left( \hat{\mu}_{i,t_s:t'} + S_{i,t_s:t'} < \hat{\mu}_{i,t'+1:t_p} - S_{i,t'+1:t_p} \right)$ or $\left( \hat{\mu}_{i,t_s:t'} - S_{i,t_s:t'} > \hat{\mu}_{i,t'+1:t_p} + S_{i,t'+1:t_p} \right)$ **then**
4:            Return True
5:         **end if**
6:      **end for**
7: **end for**

---

We introduce the phase-based ImpCPD in Algorithm 3 in Appendix D. ImpCPD calls upon the changepoint detector CPDI only at the end of phases and saves upon computation time without incurring additional regret.

**Running time of algorithms:** UCBL-CPD (like CD-UCB, and CUSUM) calls the changepoint detection at every timestep, and ImpCPD calls upon the sub-routine only at end of phases. Hence, for a fixed horizon $T$, $K$ arms, UCBL-CPD calls the changepoint detection subroutine $O(KT)$ times while ImpCPD calls the changepoint detection $O(K \log T)$ times, thereby substantially reducing the costly operation of calculating the changepoint detection statistics. By designing ImpCPD carefully and appropriately modifying the confidence interval, this reduction comes at no additional cost in the order of regret (see Discussion 7)

## 5. Main Results

**Lemma 3.** *(Control of bad-event by Laplace method) Let, $N_{i,t_s:t}$ be the number of times an arm $i$ is pulled from $t_s$ till the $t$-th timestep such that $t > t_g$, then at the $t$-th timestep for all $\delta \in (0, \frac{1}{2}]$ it holds that,*

$$\mathbb{P}\{\xi_{i,t}^{chg}\} \leq 4\delta$$

*where the event* $\xi_{i,t}^{chg} = \left\{ \exists t' \in [t_s, t] : \left( \hat{\mu}_{i,t_s:t'} - S_{i,t_s:t'} > \hat{\mu}_{i,t'+1:t} + S_{i,t'+1:t} \right) \bigcup \left( \hat{\mu}_{i,t_s:t'} + S_{i,t_s:t'} < \hat{\mu}_{i,t'+1:t} - S_{i,t'+1:t} \right) \right\}$

*and* $S_{i,t_s:t'} = \sqrt{\left(1 + \frac{1}{N_{i,t_s:t'}}\right) \frac{\log(\sqrt{N_{i,t_s:t'} + 1/\delta})}{2N_{i,t_s:t'}}}$.

**Proof 2. (Outline)** We use the sub-Gaussian property of the bounded random variables to define a non-negative super-martingale $M_t^\lambda$. We show that it is well defined and introduce a new stopped version $M_\tau^\lambda$. By Fatou's Lemma we show that it is bounded as well. Finally, we introduce an auxiliary variable $\Lambda$ independent of all other variables and use it to control $M_\tau^\Lambda$. We use Markov's inequality to bound the probability of the event $\xi_{i,t}^{chg}$ using $M_\tau^\Lambda$. The proof is in Appendix E.

**Remark 1.** Choosing $\delta = \frac{1}{t}$, in Lemma 3 we can show that $\mathbb{P}\{\xi_{i,t}^{chg}\} \leq \frac{2}{t}$, where the event $\xi_{i,t}^{chg}$ is the bad event. Note, that $\delta$ does not depend on the knowledge of horizon $T$.

**Discussion 2.** Time-uniform bounds depends only on the number of pulls $N_{i,t_s:t'}$ and implicitly on $t$ based on the parameter $\delta$. The concentration bounds based on peeling and union bound method depends explicitly on $t$ and has larger coefficients attached to them. In Table 3 we give a comparison over the three concentration bound method involving Union bound, Peeling and Laplace method. We provide the proof of the construction of concentration bound for our changepoint detection strategy by union bound in Lemma 4 in Appendix F for completeness. The $\log \log(t)$ scaling of the peeling method is not better than the one derived by the Laplace method, unless for huge timestep $t$ ($t > 10^6$, for $\delta = 0.05$ and any $\alpha > 1$). Laplace method uses the sub-Gaussian nature of the variables to give such sharp concentration bounds as opposed to other methods.

**Theorem 1.** *(Gap-dependent bound of UCBL-CPD ) For $\eta \geq \frac{6}{2\log t + 1}$, $\delta = \frac{1}{t}$, the expected cumulative regret of UCBL-CPD using the CPD is given by,*

$$\mathbb{E}_\pi[R_t] \leq \sum_{g=1}^{G} \left\{ \underbrace{\sum_{i=1}^{K} \frac{6 \log t}{\Delta_{i,g}^{opt}}}_{(a)} + \sum_{A_g^{chg}} \left( \underbrace{\frac{18H_{2,g} \log t}{\Delta(t_g, \delta)}}_{(b)} + \underbrace{\frac{18KH_{2,g} \log t}{\Delta(t_g, \delta)}}_{(c)} \right) \right\}.$$

**Proof 3. (Outline)** For each $g \in \mathcal{G}$ we bound the probability of the bad event of discarding the optimal arm between changepoints $g - 1$ and $g$ and not detecting the

*Table 3.* Comparison of Union, Peeling and Laplace method

| Method | Confidence interval | Uniform over $t$ |
|--------|---------------------|------------------|
| Union | $\sqrt{\frac{\log(4t^2/\delta)}{2N_{i,t_s:t'}}}$ | No |
| Peeling | $\sqrt{\frac{\alpha}{N_{i,t_s:t'}}\log\left(\lceil\frac{\log(t)}{\alpha}\rceil\frac{1}{\delta}\right)}, \alpha > 1$ | No |
| Laplace | $\sqrt{\left(1+\frac{1}{N_{i,t_s:t'}}\right)\frac{\log(\sqrt{N_{i,t_s:t'}+1}/\delta)}{2N_{i,t_s:t'}}}$ | Yes |

changepoint using the Laplace method (see Lemma 3). The proof of Theorem 1 is given in Appendix G. Note, that for a changepoint $g$, compounded detection delay $d_\pi(t_g - t_{g-1}) \geq d_{\pi^*}(t_g - t_{g-1})$ as it lacks observations from $t_g$ itself. But it suffices to show that atleast $d_\pi(t_g - \tau_{g-1}) = d_\pi(t_g - (t_{g-1} + d_\pi(t_{g-1} - \tau_{g-2})))$ will not be much greater than $d_{\pi^*}(t_g - t_{g-1})$ and for each $g \in \mathcal{G}$ it is bounded with high probability as long as the $\eta$ separation is maintained by Assumption 2 (see Step 6).

**Discussion 3.** In Theorem 1, (a) is the regret suffered for finding the optimal arm between changepoints $g-1$ to $g$, (b) is the maximal regret for delayed detection of $g$, and (c) is the regret suffered for total compounded delayed detection.

**Theorem 2.** *(Gap-dependent bound of ImpCPD )* For $\eta \geq \frac{8}{2\log T+1}$, $\delta = \frac{1}{T}$, the expected cumulative regret of ImpCPD using CPDI is upper bounded by,

$$\mathbb{E}_\pi[R_T] \leq \sum_{j=1}^{G}\sum_{i\in\mathcal{A}'}\left[\underbrace{\frac{48KC_1(\gamma)\Delta_{i,g}^{opt}\log(\frac{T}{K\sqrt{\log K}})}{(K\log K)^{-\frac{3}{2}}}}_{(a)} + \right.$$

$$\left.\underbrace{\frac{16\log(\frac{T(\Delta_{i,g}^{opt})^2}{K\sqrt{\log K}})}{(\Delta_{i,g}^{opt})}}_{(b)}\right] + \sum_{j=1}^{G}\sum_{i\in\mathcal{A}_g^{chg}}\left[\underbrace{\frac{16H_{2,g}\log(\frac{T(\Delta_{i,g}^{chg})^2}{K\sqrt{\log K}})}{(\Delta_{i,g}^{chg})}}_{(c)}\right]$$

$$+\sum_{j=1}^{G}\sum_{i\in\mathcal{A}_g^{chg}}\left[\underbrace{\frac{16KH_{2,g}\log(\frac{T(\Delta_{i,g}^{chg})^2}{K\sqrt{\log K}})}{(\Delta(t_g,\delta))}}_{(d)}\right]$$

where $\gamma$ is exploration parameter, $C_1(\gamma) = \left(\frac{1+\gamma}{\gamma}\right)^4$, and $\mathcal{A}' = \{i \in \mathcal{A} : \Delta_{i,g}^{opt} \geq \sqrt{\frac{e}{T}}, \Delta_{i,g}^{chg} \geq \sqrt{\frac{e}{T}}, \forall g \in \mathcal{G}\}$.

**Proof 4. (Outline)** The key to proving this theorem is to carefully construct each geometrically increasing phase length $\ell_m$ so that the probability not pulling the optimal arm between two changepoints $g-1$ to $g$ is bounded. Simultaneously, we use the phase length $\ell_m$, confidence interval $S_{i,t_s:t_p}$ and exploration factor $\gamma$ to control the bad event of not detecting the changepoint $g$. We use Chernoff-Hoeffding

inequality to bound the probability of the bad events. We have to further balance $\ell_m$, and $S_{i,t_s:t_p}$ by carefully defining $\psi$ so that $\gamma$ is small enough and CPDI is called more often. We also have to use additional union bounds to control the event that arms are getting pulled unequal number of times within each phase length. The proof is in Appendix H.

**Discussion 4.** In Theorem 2, (a) is the regret suffered for calling the CPDI only at end of phases, (b) is the regret for finding the optimal arm between changepoints $g-1$ and $g$, (c) is the regret for delayed detection of $g$, and (d) is the regret suffered for total compounded delayed detection.

**Discussion 5.** UCBL-CPD (Thm 1) and ImpCPD (Thm 2) performance is comparable to the best detection strategy (see Lemma 2) as they have the coefficient in their compounded detection delay of order $O(K)$ that is less than order $O(\eta\log(t/\delta))$ of $C(t,\delta,\eta)$ of $\pi^*$ when $\eta$ is greater than the respective values in the theorems. This is reasonable as in the bandit setup each arm $i \in \mathcal{A}$ might be pulled a logarithmic number of times before detecting a changepoint.

**Corollary 1.** *(Gap-independent bound of UCBL-CPD )* *In the specific scenario, when all the gaps are same, that is* $\Delta_{i,g}^{opt} = \Delta_{i,g}^{chg} = \Delta(t_g,\delta) = \sqrt{\frac{K\log(T/G)}{T/G}}, \forall i \in \mathcal{A}, \forall g \in \mathcal{G}$ *and* $\delta = \frac{1}{t}$ *then the worst case gap-independent regret bound of UCBL-CPD is given by,*

$$\mathbb{E}_\pi[R_T] \leq O(\sqrt{GT}\log T).$$

**Proof 5.** The proof of Corollary 1 is given in Appendix I.

**Discussion 6.** In Corollary 1, the largest contributing factor to the gap-independent regret of UCBL-CPD is of the order $O\left(\sqrt{GT}\log T\right)$, same as that of DUCB but weaker than CUSUM and SWUCB. The additional $O(\log T)$ factor is the cost UCBL-CPD must pay for not knowing $T$ and $G$.

**Corollary 2.** *(Gap-independent bound of ImpCPD )* *In the specific scenario, when all the gaps are same, that is* $\Delta_{i,g}^{opt} = \Delta_{i,g}^{chg} = \Delta(t_g,\delta) = \sqrt{\frac{K\log(T/G)}{T/G}}, \forall i \in \mathcal{A}, \forall g \in \mathcal{G}$ *and setting* $\delta = \frac{1}{T}, \gamma = 0.05$ *then the worst case gap-independent regret bound of ImpCPD is given by,*

$$\mathbb{E}_\pi[R_T] \leq C_1 G^{1.5} K^{4.5}(\log K)^2 + O(\sqrt{GT})$$

*where* $C_1$ *is an integer constant.*

**Proof 6.** The proof of Corollary 2 is given in Appendix J.

**Discussion 7.** In Corollary 2, the largest contributing factor to the gap-independent regret of ImpCPD is of the order $O\left(\sqrt{GT}\right)$. This is lower than the regret upper bound of DUCB, SWUCB, EXP3.R and CUSUM (Table 1). The smaller the value of the exploration parameter $\gamma$ the larger is the constant $C_1$ associated with the factor $GK^{4.5}(\log K)^2$. Now, $\gamma$ determines how frequently CPDI is called by ImpCPD and by modifying the confidence interval and phase-length we have been able to control the probability of not

detecting the changepoint at the cost of additional regret that only scales with $K$ and not with $T$.

**Theorem 3.** *(Lower Bounds for oracle policy) The lower bound of an oracle policy $\pi^*$ for a horizon $T$, $K$ arms and $G$ changepoints is given by,*

$$\mathbb{E}_{\pi^*}[R_T] \geq \min\left\{\Omega\left(\sum_{g=1}^{G}\sum_{i=1}^{K}\frac{\log\left(T/(GH_{1,g})\right)}{\Delta_{i,g}^{opt}}\right), \Omega\left(\sqrt{GT}\right)\right\}$$

*where, $H_{1,g} = \sum_{i=1}^{K}\frac{1}{(\Delta_{i,g}^{opt})^2}$ is the hardness of the problem.*

**Proof 7.** The key observation to prove this theorem is that the worst case scenario can occur when environment changes uniform randomly. A similar argument has also been made in the adaptive-bandit setting of Maillard & Munos (2011). So, let the horizon $T$ be divided into $G$ slices, each of length $(T/G)$. For each of these slices an oracle algorithm using OCUCB (Lattimore, 2015) should get the optimal SMAB regret without suffering any delay. The proof is in Appendix K.

**Discussion 8.** This lower bound is weaker than the bound proposed in Wei et al. (2016) as they do not require the knowledge of $G$ or $T$. But we provide this for completion to discuss oracle-based bounds and also because the previous approaches do not touch upon this approach. For a non-oracle policy the additional trade-off between the changepoint gap and the next optimality gap is captured by $H_{2,g}$. As long as the delayed detection is bounded with high probability we should get a similar scaling for a good detection algorithm minimizing regret for each of these slices of length $(T/G)$. ImpCPD which has a gap-independent regret upper bound of $O(\sqrt{GT})$ reaches the lower bound of the policy $\pi^*$ in an order optimal sense. Also, in the special case when all the gaps are same such that for all $i \in \mathcal{A}, g \in \mathcal{G}$, $\Delta_{i,g}^{opt} = \Delta(t_g, \delta) = \Delta_{i,g}^{chg}$, $H_{1,g} = K(\Delta(t_g, \delta))^{-2}$ and $H_{2,g} = 1$, then ImpCPD with a gap-dependent bound of $O\left(\sum_{g=1}^{G}\sum_{i=1}^{K}\frac{\log(T/H_{1,g})}{\Delta_{i,g}^{opt}}\right)$ matches the gap dependent lower bound of $\pi^*$ except the factor $G$ in the log term (ignoring the $\log(\frac{1}{\sqrt{\log K}})$ term).

# 6. Experiments

We compare UCBL-CPD and ImpCPD against Oracle Thompson Sampling (OTS), EXP3.R, Discounted Thompson Sampling (DTS), Discounted UCB (DUCB), Sliding Window UCB (SWUCB), Monitored-UCB (M-UCB) and CUSUM-UCB (CUSUM) in four environments . The oracle algorithms have access to the exact changepoints and are restarted at those changepoints. For each of the experiments, we average the performance of all the algorithms over 100 independent runs. More discussions on parameter selection for the algorithms can be found in Appendix M and additional experiments are shown in Appendix N.

**Experiment 1 (Bernoulli 3 arms):** This experiment is conducted to test the performance of algorithms in Bernoulli distribution over a short horizon $T = 4000$ and and small number of arms $K = 3$. There are 3 changepoints in this testbed and the expected mean of the arms changes as shown in equation 2. The experiment is shown in Figure 2(a) where we can clearly see that UCBL-CPD and ImpCPD detect the changepoints at $t = 1000$ and $t = 2000$ with a small delay and restarts. However, because of the small changepoint gap at $t = 3000$ it takes some time to adapt and restart. UCBL-CPD and ImpCPD perform better than all the passively adaptive algorithms like DTS, DUCB, SWUCB, and actively adaptive algorithm like EXP3.R, CUSUM and is only outperformed by OUCB1 and OTS which have access to the oracle. The performance of UCBL-CPD is similar to ImpCPD in this small testbed. Because of the short horizon and a small number of arms, the adaptive algorithms CUSUM and EXP3.R are outperformed by passive algorithms DUCB, SWUCB, and DTS.

$$\begin{aligned} r_1 = 0.1, r_2 = 0.2, r_3 = 0.9 \quad &\textbf{if } t = [1, 1000]; \\ r_1 = 0.4, r_2 = 0.9, r_3 = 0.1 \quad &\textbf{if } t = [1001, 2000]; \\ r_1 = 0.5, r_2 = 0.1, r_3 = 0.2 \quad &\textbf{if } t = [2001, 3000]; \\ r_1 = 0.2, r_2 = 0.2, r_3 = 0.3 \quad &\textbf{if } t = [3001, 4000]. \quad (2) \end{aligned}$$

**Experiment 2 (Jester dataset):** This experiment is conducted to test the performance of algorithms when our model assumptions are violated. We evaluate on the Jester dataset (Goldberg et al., 2001) which consist of over 4.1 million continuous ratings of 100 jokes from 73,421 users collected over 5 years. We use Jester because there exist a high number of users who have rated all the jokes, and so we do not have to use any matrix completion algorithms to fill the rating matrix. The goal of the learner is to suggest the best joke when a new user comes to the system. We consider 20 users who have rated all the 100 jokes and use SVD to get a low rank approximation of this rating matrix. Most of the users belong to three classes who prefer either joke number 49, 88, or 90. We uniform randomly sample 2 users from each of the 3 classes (49, 88, 90). Then we divide the horizon $T = 150000$ into 6 changepoints starting from $t = 1$ and at an interval of 25000 we introduce a new user from one of the three classes in round-robin fashion starting from users who prefer joke 49. We change the user at changepoints to simulate the change of distributions of arms and hence a single learning algorithm has to adapt multiple times to learn the best joke for each user. A real-life motivation of doing this may stem from the fact that running an independent bandit algorithm for each user is a costly affair and when users are coming uniform randomly a single algorithm may learn quicker across users if all the users prefer a few common items. Note, that we violate Assumption 2 and Assumption 3 because the horizon is small, the number of arms is large and gaps are too small to be

detectable with sufficient delay. In Figure 2(b) we see that ImpCPD outperforms most of the other algorithms except OTS . ImpCPD and UCBL-CPD is only able to detect 2 of the 6 changepoints and restart while CUSUM failed to detect any of the changepoints. Note that UCBL-CPD and ImpCPD performs slightly worse than DUCB in this testbed. This shows the importance of Assumption 2 and 3, that is when gaps are small, and changepoints are less separated, *all* the change-point detection techniques will perform badly in those regimes.

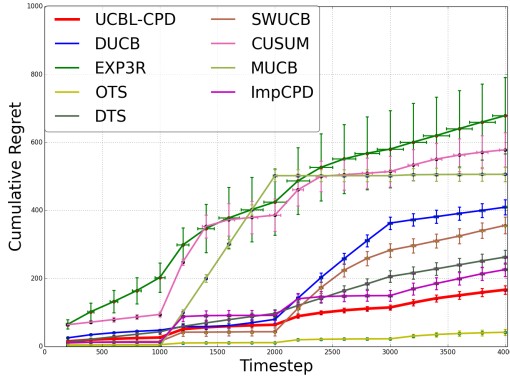

(a) Expt-1: 3 Bernoulli-distributed arms.

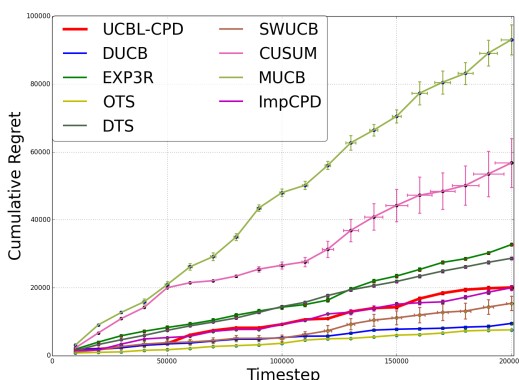

(b) Expt-2: 20 Users, 100 items, Rank 3 approximation of Jester Dataset

*Figure 2.* Cumulative regret of algorithms in Jester dataset

## 7. Related Works

Previous algorithms can be broadly divided into passive and actively adaptive algorithms. Passive algorithms like Discounted UCB (DUCB) (Kocsis & Szepesvári, 2006), Sliding Window UCB (SWUCB) (Garivier & Moulines, 2011) and Discounted Thompson Sampling (DTS) (Raj & Kalyani, 2017) do not actively try to detect changepoints and thus perform badly when changepoints are of large magnitude and are well-separated. The actively adaptive algorithm EXP3.R (Allesiardo et al., 2017) is an adaptive alternative to EXP3.S (Auer et al., 2002b) which was proposed for

arbitrary changing environments. But EXP3.R is primarily intended for adversarial environments and thus is conservative when applied to a piecewise i.i.d. environment. The recently introduced actively adaptive algorithms CD-UCB (Liu et al., 2017), CUSUM (Liu et al., 2017) and M-UCB (Cao et al., 2018) rely on additional forced exploration for changepoint detection. With $\alpha$ probability they employ some changepoint detection mechanism or pull the arm with highest UCB with $(1 - \alpha)$ probability (exploitation). This $\alpha$ is hard to tune in experiments and come with limited theoretical guarantees. CD-UCB requires that the exploration parameter is set to $\alpha = 0$ for proving theoretical guarantees. CUSUM (Liu et al., 2017) performs a two-sided CUSUM test to detect changepoints and it empirically outperforms CD-UCB. CUSUM requires the knowledge of $G$ and $T$ for tuning $\alpha$ and its theoretical guarantees only hold for Bernoulli rewards for widely separated changepoints. Also, CUSUM wrongly applies Hoeffding inequality to a random number of pulls (see eq (31), (32) in Liu et al. (2017)) which raises serious concerns about the validity of the rest of their analysis. Finally, M-UCB also requires the knowledge of $G$ and $T$ for theoretical guarantees. Another approach involves the Generalized Likelihood Ratio Test which was recently studied by Maillard (2019) and Besson & Kaufmann (2019). This is a different approach than others and looks at the ratio of the likelihood of the sequence of rewards coming from two different distributions and calculates the sufficient statistics to detect changepoints. An extended discussion can be found in Appendix A.

## 8. Conclusions and Future Works

We studied the piecewise i.i.d environment under Assumption 1, 2 and 3 such that actively adaptive algorithms do not need to conduct forced exploration to detect changepoints. We studied two UCB algorithms, UCBL-CPD and ImpCPD which are adaptive and restarts once the changepoints are detected. We derived the first gap-dependent bounds for these actively adaptive algorithms incorporating the hardness factor. The anytime UCBL-CPD uses the Laplace method to derive sharp concentration bound, and ImpCPD achieves the order optimal regret bound which is an improvement over all the existing algorithms (in a specific setting). Empirically, they perform very well in various environments and is only outperformed by oracle algorithms. Future works include incorporating the knowledge of localization in these adaptive algorithms.

### 8.1. Acknowledgments

This work has been supported by CPER Nord-Pas de Calais/FEDER DATA Advanced data science and technologies 2015-2020, the French Ministry of Higher Education and Research, Inria Lille Nord Europe, CRIStAL, and the French Agence Nationale de la Recherche (ANR), under grant ANR-16-CE40-0002 (project BADASS).

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
