# OpenReview forum: "Distribution-dependent and Time-uniform Bounds for Piecewise i.i.d Bandits"
_ICML.cc/2019/Workshop/RL4RealLife — RL4RealLife 2019_

### Official Review · AnonReviewer2 · 2019-05-25
**In summary, I suggest a ‘weak reject’.**

**Rating:** 2
**Confidence:** 5

**Review:**

This paper study the piece-wise i.i.d environment, also known as the piecewise stationary environment, for multi-armed bandit learning. Under the considered environment, they proposed two UCB-based bandit solutions with change modules.

Positive aspects of the paper:
—They studied the distribution-depend bounds, proved the gap-dependent bound (upper and lower bound) of their proposed solution, and also proved the time-uniform bound of a variant of their proposed solution. The proof of gap-dependent and the proof of time-uniform bound are of independent interest.
-- The proposed solutions sounds valid and reasonable.

Negative aspects of the paper:
— The paper is not well written, on one hand, there are many typos in the paper; on the other hand, their solution is not well motivated and many statements are not clearly explained. For example, the following sentences do not make sense to me.

‘However, we can do better by utilizing three assumptions that the previous approaches take and reduce the regret incurred
for forced explorations. These are assumption on global changepoint (1), a different separation of changepoint assumption (2) based on maximal delay, and a slightly stronger assumption (3) involving minimum detectable gaps ((tg; )).’

— The empirical study is too weak: The bernoulli 3 arms experiment is just a very simple numerical study. And for the Jester experiment, the experimental results are basically negative because the gaps are too small. I think it will be more helpful if the authors can try to control the gap (for example vary the gap) to see how it will affect their proposed algorithm and the other baselines.

— There are some over claim in the paper. For example, the authors emphasized that no need for forced exploration is needed for their algorithm (under their required assumptions). However, some of existing work, such as M-UCB, CUSUM,  in fact can already do this.

---

### Official Review · AnonReviewer1 · 2019-05-27
**Solid contribution to bandits with changepoints**

**Rating:** 5
**Confidence:** 3

**Review:**

The paper proposes two algorithms for multi-armed bandits where the arms have i.i.d. stochastic rewards but their distribution can change at specific changepoints which are unknown to the algorithm but have to be detected. While one algorithm assumes knowledge of the time-horizon T, the other is anytime. Under assumptions on the changepoints (separation and detectability), the paper provides gap-dependent and gap-independent regret bounds. By comparing these to a lower-bound derived for an oracle that knowns the changepoints, the paper concludes that one of the algorithms achieves order-optimal gap-independent regret. Finally, both algorithms are empirically compared against competitors on two benchmarks.

The derived performance bounds are insightful and to the best of my knowledge novel. I particularly enjoyed the clear presentation of the time-uniform confidence bounds based on LaPlace method and how they enable an anytime algorithm. The paper is quite dense but this is somewhat unavoidable due to the technical nature of the contributions. Nonetheless, the explanation and motivation of Assumption 2 can be improved. I suggest making better use of Figure 1 by actually discussing it in the text or a caption. The theoretical treatment is rigorous and the result significant. The experiments indicate that the methods performs well compared to competitors but to better assess the usefulness of both proposed methods, it would be great to see a more comprehensive comparison on benchmarks where the (somewhat unrealistic) assumptions are violated (besides the Jester Dataset). That said, I do understand that the primary contributions of this paper are theoretical nature and therefore believe that such a more thorough experimental comparison can be left as future work.

Detailed / minor comments:
- The formulation of Corollary 2 is somewhat ambiguous. From the statement, It can be understood that the uniform setting for all gaps is a condition under which the regret bound hold.  The fact that this is the settings where this bound is somewhat tight but holds for any gaps is only clear from context. I suggest rewording to avoid this potential confusion.
- Footnote 1 from Table 1 seems to be missing?
- Figure 1 quite complicated but not explained sufficiently in the text / caption.
- Different methods in Figure 2 hard to distinguish even on color prints
- Why is there an indicator in the regret definition R__T? Doesn't this mean that if the algorithm plays the optimal arm every time, the regret still grows with the optimal arm mean?

---

### Decision · Program_Chairs · 2019-05-28

Accept